# Exploring the Use of Fitbit Consumer Activity Trackers to Support Active Lifestyles in Adults with Type 2 Diabetes: A Mixed-Methods Study

**DOI:** 10.3390/ijerph182111598

**Published:** 2021-11-04

**Authors:** William Hodgson, Alison Kirk, Marilyn Lennon, Gregor Paxton

**Affiliations:** 1Physical Activity for Health, Department of Psychological Sciences and Health, Faculty of Humanities and Social Sciences, University of Strathclyde, Glasgow G1 1XQ, UK; alison.kirk@strath.ac.uk (A.K.); gregor.paxton.2016@uni.strath.ac.uk (G.P.); 2Digital Health and Wellness, Department of Computer and Information Sciences, Faculty of Science, University of Strathclyde, Glasgow G1 1XQ, UK; marilyn.lennon@strath.ac.uk

**Keywords:** type 2 diabetes, Fitbit, physical activity, activity tracker

## Abstract

Background: People with type 2 diabetes are less active than those without the condition. Physical activity promotion within diabetes health care is limited. This project explored the use of Fitbit activity trackers (Fitbit, San Francisco, CA, USA) to support active lifestyles in adults with type 2 diabetes through a mixed-methods study. Methods: Two stages were conducted. In stage 1, adults with type 2 diabetes used a Fitbit Charge 4 (Fitbit, San Francisco, CA, USA) for 4 weeks. Fitbit and self-reported physical activity data was examined through quantitative analysis. Qualitative analysis was conducted to explore the experiences of participants. In stage 2, health professionals were interviewed to examine their views on using Fitbit activity trackers within type 2 diabetes care. Results: Adults with type 2 diabetes were recruited for stage 1 and adult health care and fitness professionals were recruited for stage 2. Stage 1 participants’ self-reported increases in physical activity (mean weekly minutes of walking increased from 358.75 to 507.50 min, *p* = 0.046) and a decrease in sedentary behaviour (mean daily hours of sedentary behaviour decreased from 10.65 to 10.05 h, *p* = 0.575). Fitbit activity data ranges identified individuals who led inactive and sedentary lifestyles below levels recommended and in need of physical activity support to reduce the risk to their health. During interviews, participants stated that the Fitbit activity tracker motivated them to be more active. Stage 2 participants intimated that Fitbit activity trackers could improve the promotion of physical activity within type 2 diabetes care. Interventions involving the Fitbit premium service, community prescription and combined use of Fitbits with physical activity behaviour change models were recommended by stage 2 participants. Conclusions: This study found that there is future scope for using Fitbit activity trackers to support active lifestyles in adults diagnosed with type 2 diabetes.

## 1. Introduction

Type 2 diabetes mellitus is a non-communicable disease, which occurs when blood glucose levels rise (hyperglycaemia). Symptoms of hyperglycaemia include unexplained weight loss, blurred vision, frequent urination, tiredness and thirst [1]. Clinically, diabetes is diagnosed when fasting plasma glucose is ≥7.0 mmol/L or two-hour plasma glucose after an oral glucose tolerance test is ≥11.1 mmol/L or HbA1c is ≥48 mmol/mol or random plasma glucose is ≥11.1 mmol/L [2]. Risk factors of developing type 2 diabetes include poor diet, older age, obesity, sedentary behaviour and a lack of physical activity. Health risks associated with type 2 diabetes are macrovascular (cardiovascular disease) and microvascular (retinopathy, nephropathy and neuropathy) [3]. In 2019, worldwide, an estimated 463 million people were living with type 2 diabetes. By 2045, this number is expected to rise to 700 million. Type 2 diabetes is the most common form of the disease, with 374 million people living with the condition globally. The estimated global cost of treating type 2 diabetes in 2019 was $760 billion [1]. In 2019, 3.9 million people were diagnosed with diabetes in the UK. Of these, 3.4 million were living with type 2 diabetes [4]. Diabetes is a major cause of premature death. Using Cox regression (CSH) to calculate the hazard risk of cause-specific death, researchers found that the hazard risk of cardiovascular death for men was 2.03 and 2.28 for women, cancer death for men was 1.37 and 1.68 for women and non-cardiovascular/cancer death was 1.53 for men and 1.89 for women [5].

Physical activity is described as ‘any bodily movement produced by the skeletal muscles that results in an increase over resting energy expenditure’. Physical activity can be part of work, daily living, sport and leisure-time activities [6]. An active lifestyle has been shown to reduce the risk of developing chronic non-communicable diseases such as type 2 diabetes [7]. A position statement by the American Diabetes Association notes that physical activity can help reduce the risk of developing type 2 diabetes and should form part of a health care treatment programme for those diagnosed with the disease [8]. Structured exercise can significantly reduce the blood glucose levels (HbA1c) levels of type 2 diabetes patients (weighted mean difference (WMD) −0.67%) [9]. Aerobic-based exercise (>150 min of moderate- to vigorous-intensity physical activity) has been found to significantly reduce the HbA1c levels of type 2 diabetes patients (WMD 0.86%) [10].

Studies have shown that adults with type 2 diabetes are generally less physically active and spend more time being sedentary than those without the disease. Lifestyle interventions including physical activity have been recommended to form part of the health care of type 2 diabetes patients with a view to increasing activity levels and reducing sedentary time [11].

Sedentary behaviour is defined as any waking time spent lying, reclining or sitting and involving an energy expenditure of ≤1.5 metabolic equivalents (METs). Sedentary behaviour is different from being inactive and, for those with type 2 diabetes, reduces the ability of insulin to uptake glucose from the blood into the body cells [12]. Sedentary behaviour increases the risk of developing cardiovascular disease and a greater reliance on medication to manage type 2 diabetes [13]. A high level of sedentary behaviour significantly increases the risk of developing type 2 diabetes [12]. Reducing sedentary time and engaging in even light intensity physical activity (>1.5–3 METs) can improve insulin sensitivity and reduce the risk of developing type 2 diabetes [14].

The benefits of physical activity behavioural change interventions, based on the social cognitive theory or transtheoretical model, for people diagnosed with type 2 diabetes, has been shown to increase both objectively and self-reported measures of physical activity (Standardised Mean Difference (SMD) 0.45 and 0.79, respectively) and significantly decrease Hb1Ac levels (WMD −0.32%) and BMI (−1.05 kg/m^2^) [15]. The delivery of physical activity behaviour change interventions by health care professionals treating patients with type 2 diabetes is challenging. Health care professionals have cited a lack of training, lack of time and the lack of suitable delivery programmes as barriers to the effective provision of physical activity interventions. As a solution, health care professionals have recommended a structured referral route for patients, dedicated physical activity practitioners, specific physical activity content designed for patients with type 2 diabetes and combining physical activity promotion with patient’s clinical data [16].

With the global rise in numbers of people diagnosed with type 2 diabetes, pressure on health care resources is growing and the need for effective and efficient treatments has never been greater. Online internet-based type 2 diabetes websites offer a cost-effective method of delivering educational and self-management material for patients. Such systems have been shown to increase the knowledge of patients, reduce the need to travel for treatments and reduce the feeling of isolation for patients [17]. My Diabetes My Way (MyWay, Dundee, UK) is a web-based support system for diabetes patients, which allows them access to support material and their medical records, in particular medication and blood glucose levels. Initially developed for the National Health Service (NHS) Scotland, this system has been exported for use throughout the United Kingdom and several European countries. In Scotland, >55,000 diabetes patients are registered with the system. My Diabetes My Way includes basic physical activity advice for patients though research has highlighted that this section of the package is less used than other elements [18]. In 2019, My Diabetes My Way allowed patients to upload their Fitbit activity tracker data onto the system. At present, this data is not being analysed or utilised [19].

The use of digital technology to support patients with type 2 diabetes is a growing field of research. The variety of equipment available to patients includes consumer activity trackers, pedometers, blood glucose monitors and smartphone mobile applications. The advantages of these technologies are that they can allow health care professionals to remotely monitor patients and reduce the need for patients to regularly attend clinics [20]. Research has shown that consumer activity trackers when combined with physical activity behavioural change interventions and web-based type 2 diabetes self-management programmes can reduce the blood glucose levels (Hb1Ac) of patients (8.0% ± 0.7% to 7.3% ± 0.9%) [21]. Fitbit activity trackers when used as part of a type 2 diabetes physical activity intervention can lower patients BMI, Hb1Ac levels and increase their levels of physical activity [22]. Fitbit consumer activity trackers are a valid and reliable method of measuring physical activity (steps, distance walked, energy expenditure, physical activity intensity and sedentary behaviour). When compared with laboratory-based tests of physical activity, Fitbit activity trackers have been shown to have large significant correlation coefficients of between 96.5 and 99.1 [23].

The aim of this study was to explore through quantitative and qualitative data (mixed-methods design) the use of Fitbit consumer activity trackers to support active lifestyles in adults with type 2 diabetes.

## 2. Materials and Methods

This study was granted ethical approval by the University Ethics Committee on the 14 December 2020 (UE20/77 refers). All participants provided written consent to take part. To ensure anonymity, participants were randomly allocated a unique four-digit identification number.

### 2.1. Design

A mixed-methods analysis was chosen to examine the use of Fitbit activity trackers to support an active lifestyle in people diagnosed with type 2 diabetes. Mixed-methods analysis is commonly used during health-related interventions and integrates both quantitative and qualitative analysis to provide a more in-depth multi-dimensional investigation. This study design allowed for the participants experiences of an intervention (qualitative) to be combined with study measurements (quantitative) and can assist in the future development of type 2 diabetes treatment plans, which focus on the needs of the patient. Specifically, during this study, an explanatory mixed-methods analysis was applied which implemented the quantitative element first and then the qualitative element [24]. This study was divided into two stages. In stage 1, participants were recruited to trial the use of a Fitbit Charge 4 consumer activity tracker for a period of 4 weeks and follow-up interviews were conducted to explore their experiences. In stage 2, health care and fitness professionals were interviewed to examine their views on using Fitbit activity trackers within type 2 diabetes health care. Figure 1 provides an overview of the study procedures.

### 2.2. Participants

#### 2.2.1. Stage 1

In total, 12 adults (8 males, 4 females) with type 2 diabetes participated in stage 1 of this study. The majority (92%) were white and living in an urban setting (83%) with all residing in Scotland. Table 1 shows individual profiles for each participant.

Participants were recruited through past involvement in type 2 diabetes research (via email) and social media posts (via Facebook (Meta Platforms, Menlo Park, CA, USA) and Twitter (Twitter Inc, San Francisco, CA, USA)). Recruitment inclusion criteria included adults aged 18+ years, diagnosed with type 2 diabetes, residing in the U.K., able to read and write in English and have access to the internet for the transfer of activity data. Exclusion criteria included having been advised by a health care professional not to undertake physical activity and currently using a Fitbit activity tracker. Users of alternative fitness trackers were not excluded from this study as the focus was on Fitbit devices. The study participant information sheet and consent form were uploaded onto the secure Qualtrics survey system. A link to this form was emailed to participants and their consent recorded. Once consent was received, participants were emailed a link to a baseline questionnaire on the Qualtrics system (Qualtrics XM, London, UK). This questionnaire gathered demographic, medical and physical activity characteristics of the participants. The physical activity questions were based on the International Physical Activity Questionnaire (short form). This is a valid and reliable method of measuring self-reported physical activity and sedentary behaviour [25].

#### 2.2.2. Stage 2

Participants (*n* = 7) were recruited through University contacts (via email) and through social media posts (via Facebook and Twitter). Recruitment criteria included adults aged 18+ years, residing in the UK and experience of type 2 diabetes health care or physical activity for health or fitness activity trackers. The study participant information sheet and consent form were uploaded onto the secure Qualtrics survey system. A link to this form was emailed to participants and their consent recorded. Once consent was received, a link to the baseline demographics questionnaire on the Qualtrics survey system was emailed to participants.

### 2.3. Procedure

Stage 1 participants were provided with a Fitbit Charge 4 consumer activity tracker for the 4 week quantitative element of the study. The Fitbit activity tracker was used to align with the Fitbit data presently collected on the ‘My Diabetes My Way’ platform. No other activity trackers are presently linked with this system. Participants were asked to wear the Fitbit device continually for the 4 week period. If needing recharge, participants were advised to undertake this during the night as sleep was not being recorded for this study. A dedicated study Gmail (Google LLC, Menlo Park, CA, USA) account was setup for the registration of each Fitbit Charge 4. Using this account, each activity tracker was registered on the Fitbit website by adding the participants unique identification number, i.e., by adding +P1000 to the study Gmail address prior to the @ icon. This negated the need to setup a new Gmail account for each participant during registration. Due to COVID-19 restrictions each Fitbit was posted to the participant with instructions on how to access their specific Fitbit account. Participants were provided with no further advice on how to use the device. The research team had access to each Fitbit account and collected data in relation the user’s: daily step count, weekly minutes of low (>1.5–<3 METs) and moderate to vigorous physical activity (3–>6 METs) and sedentary time. After the 4 week trial, participants returned the Fitbit activity tracker via a pre-paid postal service.

Each stage 1 participant also completed the baseline and end of study questionnaire. These two questionnaires included seven questions from the International Physical Activity Questionnaire (Short Form). These questions are a valid and reliable method of assessing the types and intensity of physical activity and sedentary time within the daily lives of participants [26]. The baseline questionnaire included questions designed to gather participant demographic details including weight, height, HbA1c levels and duration of being diagnosed with type 2 diabetes.

Follow-up one-to-one interviews were conducted with stage 1 participants to explore through qualitative analysis their use, acceptability, and experiences of using the Fitbit activity tracker to support an active lifestyle. This included users preferred Fitbit functions, motivation to be physically active, ease of use, security and data sharing, need for additional support and present physical activity promotion within diabetes health care. A semi-structured interview schedule was prepared. The topic guide included three main themes—data fusion (how Fitbit data can be used to support patients to be more active), data protection (how Fitbit data can be securely integrated into present health care systems) and data support (what additional support do patients need). Utilising abductive analysis methods, interview questions started with the main topic theme working down into the participants experiences in more detail (inductive analysis). As interviews progressed and participants provided greater detail beyond the initial question asked these were guided back towards the top-level themes (deductive analysis). Interviews were conducted over the secure University Zoom (Zoom Video Communications Inc, San Jose, USA) conference system. Interviews were designed to take between 30 and 40 min.

In stage 2, adult health care and fitness professionals, including Fitbit management employees, were recruited to explore through qualitative analysis their experiences, knowledge and feasibility of using Fitbit activity trackers to support an active lifestyle in patients diagnosed with type 2 diabetes. As in stage 1, a semi-structured interview schedule was prepared, and one-to-one interviews were conducted over the secure University Zoom conference system. Each interview was recorded using the inbuilt recording system on zoom and later transcribed verbatim. Interviews were designed to take between 30 and 40 min.

#### Analysis

One-way within-subjects analysis of variance (ANOVA) tests were conducted on each of the activity components recorded on the Fitbit device. Paired-samples *t*-tests were conducted on participants’ self-reported baseline and end of study mean weekly minutes of moderate- to vigorous-intensity physical activity and mean daily hours of sedentary time. A Wilcoxon signed ranks test was conducted on participants’ self-reported baseline and end of study mean weekly minutes of walking. SPSS 27 software (IBM Inc, New York, NY, USA) was used to conduct the statistical analysis.

All stage 1 and stage 2 interviews were recorded and later transcribed verbatim. Abductive qualitative thematic analysis was undertaken on the transcribed interviews and themes identified in relation to the participants experiences and knowledge. Abductive analysis incorporates deductive (top-down approach) and inductive (bottom-up approach) to provide a more detailed and flexible method of exploring the experiences of participants during a study [26]. NVIVO 12 software (QSR International, Melbourne, Australia) was used to undertake the thematic analysis.

## 3. Results

### 3.1. Stage 1—Adults Diagnosed with Type 2 Diabetes

### 3.2. Quantitative Analysis

#### 3.2.1. Participants Fitbit Data

One-way within-subjects ANOVAs were conducted to compare the effect of participants use of a Fitbit Charge 4 activity tracker over a four week period on mean daily steps taken (number), mean daily sedentary time (hours), mean weekly light intensity physical activity (minutes) and mean weekly moderate- to vigorous-intensity physical activity (minutes).

##### Daily Steps (Number Taken)

A test of normality was carried out and the assumption was met. Mauchly’s test of sphericity produced a significant result (*p* = 0.003). Reporting Greenhouse–Geisser showed that there was a non-significant large effect of Fitbit use on mean daily steps taken F (1.42, 14.18) = 2.36, *p* = 0.140, *n*^2^ = 0.19. Figure 2 shows that participants’ mean daily steps taken decreased from 6597.82 ± 3449.80 in week one to 6028.84 ± 2507.35 in week two to 4990.83 ± 2457.18 in week three and increased to 5601.61 ± 3156.05 in week four. Participants’ range of mean daily steps was 2185.00–11,352.00 in week one, 2143.86–9977.14 in week two, 1870.71–8266.00 in week three and 1854.29–11,024.71 in week four. Figure 3 shows the individual participants’ mean daily steps.

##### Daily Sedentary Time (Hours)

A test of normality was undertaken, and the assumption was met. Mauchly’s test of sphericity produced a significant result (*p* = 0.016). Reporting Greenhouse–Geisser showed that there was a non-significant large effect of Fitbit use on mean daily sedentary time F (1.64, 17.98) = 1.22, *p* = 0.310, *n*^2^ = 0.10. Figure 4 shows that participants’ mean daily sedentary time increased from 6.45 ± 3.76 h in week one to 7.58 ± 4.17 h in week two and decreased to 7.56 ± 4.56 h in week three and increased to 8.00 ± 3.54 h in week four. Participants’ range of mean daily sedentary time was 2.19–15.35 h in week one, 2.25–15.40 h in week two, 2.22–14.66 h in week three and 2.22–13.64 h in week four. Figure 5 shows participants’ mean daily sedentary time for each week of the study.

##### Weekly Light Intensity Physical Activity (Minutes)

A test of normality was undertaken, and the assumption was met. Mauchly’s test of sphericity produced a significant result (*p* = 0.025). Reporting Greenhouse–Geisser showed that there was a non-significant medium effect of Fitbit use on mean weekly light intensity physical activity F (1.68, 18.49) = 0.76, *p* = 0.462, *n*^2^ = 0.06. Figure 6 shows that participants’ mean weekly light intensity physical activity increased from 1128.00 ± 479.91 min in week one to 1224.58 ± 602.11 min in week two to 1281.33 ± 534.08 min in week three and decreased to 1159.75 ± 415.38 min in week four. Participants’ range of mean weekly light intensity physical activity was 272.00–2053.00 min in week one, 124.00–2502.00 min in week two, 564.00–2235.00 min in week three and 634.00–1976.00 min in week four. Figure 7 shows individual participants’ mean weekly minutes of light intensity physical activity.

##### Weekly Moderate to Vigorous Physical Activity (Minutes)

A test of normality was carried out and the assumption was met. Mauchly’s test of sphericity produced a non-significant result (*p* = 0.062). Reporting sphericity assumed showed that there was a non-significant large effect of Fitbit use on mean weekly moderate- to vigorous-intensity physical activity F (3, 33) = 2.75, *p* = 0.058, *n*^2^ = 0.20. Figure 8 shows that participants’ mean weekly moderate- to vigorous-intensity physical activity decreased from 326.42 ± 272.38 min in week one to 196.50 ± 114.31 min in week two and decreased to 179.33 ± 162.22 min in week three and increased to 216.50 ± 226.46 min in week four. Participants’ range of mean weekly moderate to vigorous physical activity was 0.00–776.00 min in week one, 16.00–349.00 min in week two, 0.00–502.00 min in week three and 3.00–770.00 min in week four. Figure 9 shows individual participants’ mean weekly minutes of moderate- to vigorous-intensity physical activity.

#### 3.2.2. Participant Baseline and End of Study Questionnaires

Paired-samples *t*-tests (normal distributed data) were conducted to compare the effect of participants use of a Fitbit Charge 4 activity tracker at baseline and at the end of the study on self-reported mean weekly minutes of moderate- to vigorous-intensity physical activity and mean daily minutes of sedentary time. A Wilcoxon Signed Ranks repeated-measure tests (non-normal distributed data) was conducted to compare the effect of participants use of a Fitbit Charge 4 activity tracker at baseline and at the end of the study on self-reported mean weekly minutes of walking time.

##### Participants’ Self-Reported Mean Weekly Minutes of Moderate- to Vigorous-Intensity Physical Activity

A repeated-measure *t*-test found that there was a non-significant small effect of Fitbit use on self-reported mean weekly minutes of moderate- to vigorous-intensity physical activity between baseline and end of study t (11) = −0.20, *p* = 0.848, Cohen’s *d* = 0.06. Figure 10 shows that mean weekly minutes of moderate- to vigorous-intensity physical activity at baseline 498.75 ± 368.03 increased to 513.33 ± 437.49 at the end of the study. At baseline the minimum number of weekly minutes was 140.00 and the maximum 1050.00. At end of study the minimum number of weekly minutes was 175.00 and the maximum 1470.00. Figure 11 shows individual participants’ mean weekly minutes of moderate- to vigorous-intensity physical activity.

##### Participants’ Self-Reported Mean Daily Hours of Sedentary Time

A repeated-measure *t*-test found that there was a non-significant small effect of Fitbit use on self-reported mean daily hours of sedentary time between baseline and end of study t (11) = −0.58, *p* = 0.575, Cohen’s *d* = 0.17. Figure 12 shows that mean daily hours of sedentary time at baseline 10.65 ± 2.53 decreased to 10.05 ± 2.25 at the end of the study. Figure 13 shows individual participants’ self-reported mean daily hours of sedentary time at baseline and end of study.

##### Participants’ Self-Reported Mean Weekly Minutes of Walking at Baseline and End of Study

A Wilcoxon Signed Ranks test found that there was a significant large effect of Fitbit use on self-reported mean weekly minutes of walking at baseline and end of study T = 1, *p* = 0.046, *r* = 0.58. Figure 14 shows that participants’ mean self-reported weekly minutes of walking at baseline 358.75 ± 242.09 increased to 507.50 ± 256.55 at the end of the study. At baseline the minimum number of weekly minutes of walking was 105.00 and the maximum 840.00. At end of study the minimum number of minutes was 105.00 and the maximum 840.00. Figure 15 shows individual participants’ mean weekly minutes of walking.

### 3.3. Qualitative Analysis

Abductive thematic analysis identified 40 sub-themes and 7 main themes. The 7 main themes were: current delivery of physical activity advice within type 2 diabetes health care, integrated elements of type 2 diabetes health care, data security and management, barriers to Fitbit use, personalisation of type 2 diabetes physical activity support, use of Fitbit as a motivational and goal setting tool and users preferred Fitbit functions. Table 2 provides a summary of the associated links between the main themes and sub-themes.

The 7 main themes were supported by in-text references from the transcribed interviews. Table 3 shows the detailed analysis of the main themes.

### 3.4. Stage 2—Health Care and Fitness Professionals

#### 3.4.1. Participants Baseline Demographic Data

In total 7 adults (3 male, 4 female) participated in stage 2 of this study. The majority (86%) were white with 71% residing in Scotland and 29% residing in England. Mean age was 44.57 ± 7.64 with range 21 -58 years. Participant’s occupations were Medical Doctor (1), Nurse (general practice) (2), Nurse (diabetes specialist) (1), Diabetes Academic (1) and Activity Tracker Professional (2).

#### 3.4.2. Qualitative Analysis

Abductive thematic analysis identified 32 sub-themes and 6 main themes relating to the use of Fitbit activity trackers to support active lifestyles in adults with type 2 diabetes. The 6 identified main themes were: present promotion of physical activity within type 2 diabetes health care, data security and management, Fitbit functionality, Fitbit health care barriers, future use of Fitbit within type 2 diabetes health care and improving physical activity promotion. Table 4 provides a summary of the associated links between the main themes and sub-themes.

The 6 main themes were supported by in-text references from the transcribed interviews. Table 5 shows the detailed analysis of stage 2 main themes.

## 4. Discussion

The aim of this study was to explore the use of Fitbit consumer activity trackers to support active lifestyles in adults with Type 2 Diabetes through a mixed-methods analysis. In a recent study by Diaz et al., 2021 the combination of quantitative and qualitative data through a mixed-methods design was found to develop a better understanding of the useability and acceptability of a newly developed mobile application. The aim of this application was to improve users knowledge in relation to the risks of cardiovascular disease [27].

Stage 1 participants produced three streams of data for analysis. These included their objective Fitbit measurements (use of Fitbit activity tracker), subjective responses to the participant questionnaires (baseline and end of study) and qualitative interviews (experiences of using Fitbit activity tracker). Though the main focus of the quantitative analysis was the variance of participants’ means for each activity the minimum and maximum figures show the disparity of participants activity levels. In general participants recording the maximum figures were achieving the UK recommended levels of physical activity. Those recording the minimum figures were failing to reach levels recommended for an active and healthy lifestyle. As such these individuals would gain most from physical activity interventions.

### 4.1. Stage 1—Adults Diagnosed with Type 2 Diabetes

#### 4.1.1. Quantitative Analysis

Participants were provided with a Fitbit Charge 4 activity tracker or use over a 4 week period. Apart from initial registration advice users were offered no further support. The Fitbit operating system requires users to have an internet connection for the transfer of data. Though all participants had an internet connection any future use of such devices needs to take such access into consideration and the potential barrier to use by some patients. Analysis of the participants Fitbit data found that activity levels in relation to daily steps and moderate- to vigorous-intensity physical activity reduced non-significantly between week 1 and week 4. It could be suggested that the Fitbit activity tracker initially motivated users to be active during the first week of wearing the device but without further support activity levels then declined. Further support in the form of a physical activity behaviour change intervention could bridge this gap as per the study by Lim et al. (2016) [20] and in particular for those recording low levels of activity. In the case of sedentary time and low intensity physical activity levels increased non-significantly between week 1 and week 4. The sedentary time increase corresponded with the decrease in activity levels apart from low intensity. The increase in low intensity physical activity is encouraging as this has been shown to improve insulin sensitivity in people diagnosed with type 2 diabetes as per the study by Sardinha, Magalhães, Santos, and Júdice (2017) [14]. In relation to the UK physical activity guidelines the participants’ mean weekly moderate- to vigorous-intensity time for each week of the trial was above the recommended 150 min though mean daily steps was below the recommended 10,000 steps for the general adult population. The data range indicated that some participants were consistently failing to achieve this recommendation while others were exceeding it (Department of Health and Social Care, 2019) [6]. Providing physical activity support for those at the lower end of the data range should be a priority for health care providers. The participants’ mean self-reported data obtained from the baseline and end of study questionnaire found that weekly minutes walking increased significantly. Weekly minutes of moderate- to vigorous-intensity physical activity increased non-significantly. Daily hours of sedentary behaviour decreased non-significantly. Self-reported physical activity data indicates that participants perceived that they did make positive changes in their physical activity and sedentary behaviour as a result of wearing the Fitbit activity tracker. This was in contradiction to the activity data gathered from the Fitbit devices.

#### 4.1.2. Qualitative Analysis

The semi-structured interviews with participants and follow-up thematic analysis identified that physical activity promotion is limited within present type 2 diabetes health care due mainly to time constraints. Participants particularly desired a more personalised health care support service focusing on their physical activity needs and lifestyle. Personalised physical activity behaviour change interventions have been shown to increase activity and decrease blood glucose levels in patients diagnosed with type 2 diabetes [15]. In contrast to the quantitative data participants stated that the Fitbit Charge 4 activity tracker had motivated them to be more physically active though the majority expressed a need for more support and analysis via health care clinicians. The step counting function on the Fitbit proved to be the most popular with participants as it provided them with a simple indication of their daily activity. Further support and advice would allow users to understand how better to use all Fitbit functions. All participants indicated that they would be happy to share their Fitbit data with health care staff and NHS-based information technology.

### 4.2. Stage 2—Health Care and Fitness Professionals

#### Qualitative Analysis

In relation to present physical activity promotion within type 2 diabetes health care participants stated that this was limited and generally only briefly discussed during consultations. The main focus of medical care was on medication, blood glucose measurements and nutritional advice. Limited time was highlighted as the main reason health care staff spent less time promoting physical activity. Participants recommended the use of dedicated trained staff as a solution to improve the promotion of physical activity. Such trained staff could focus on physical activity behaviour change programmes and direct patients to exercise support groups or facilities. These results mirrored similar findings identified in the study by Matthews, Kirk and Mutrie (2014) [16]. Participants made a number of suggestions for using a Fitbit activity tracker to support type 2 diabetes patients in respect of physical activity. The main recommendations focused on the social prescription of Fitbits through community hubs such as pharmacies, the use of the Fitbit communities function for group support and use of the Fitbit premium service, which is based on physical activity behaviour change interventions. The development of Fitbit data analysis software was also suggested as a supporting tool for both patients and health care staff. The funding of Fitbit prescription was highlighted as a potential barrier for use within the NHS.

### 4.3. Study Limitations

The sample size for this study was small reducing the power of the quantitative tests. G-Power software, considering mean variance and effect size, suggested a more appropriate sample size would be 40 participants. Pre-study objectively measured levels of physical activity were not conducted making it difficult to compare the effect of the Fitbit activity tracker between pre-trial and post-trial. During the Fitbit registration process a single email address was setup by the research team and all motivational and progress emails for each participants device were sent to this point of contact. Participants did not receive these emails which deprived them of this supportive element of the Fitbit system. During weeks 2 and 3 of the Fitbit trial the weather in the UK was particularly cold with heavy snow. This combined with the UK COVID-19 restrictions made it difficult for participants to exercise outdoors and all indoor sports facilities were closed. Extending the Fitbit trial beyond the 4 week period would have been useful and future research should consider this. Using Fitbit employees during the qualitative element of this study could have introduced product bias into the interviews.

## 5. Conclusions

The aim of this study was to explore the use of Fitbit consumer activity trackers to support active lifestyles in adults with Type 2 Diabetes through a mixed-methods design. This small study identified present limitations in the promotion of physical activity within type 2 diabetes health care. During the Fitbit trial participants increased their levels of light intensity physical activity between week 1 and week 4 (non-significant). Fitbit data identified participants who were not achieving the UK recommended moderate- to vigorous-intensity physical activity guidelines indicating that these individuals would benefit from physical activity support. Self-reported physical activity levels increased over the 4 week period and sedentary behaviour decreased indicating that participants perceived that the Fitbit supported an active lifestyle. Qualitative analysis found that users thought the Fitbit device motivated them to be more active. Health care and fitness professionals identified ways in which a Fitbit activity tracker could be used to support the promotion of physical activity within type 2 diabetes health care. Overall, there was evidence that Fitbit activity trackers could support active lifestyles in adults with type 2 diabetes though further research is suggested to identify the best methods. Such research should include measuring participant activity at baseline and extend the device trial element to allow for external factors such as weather. More detailed discussion with health care professionals could identify methods of integrating activity trackers into the care of patients.

## Figures and Tables

**Figure 1 ijerph-18-11598-f001:**
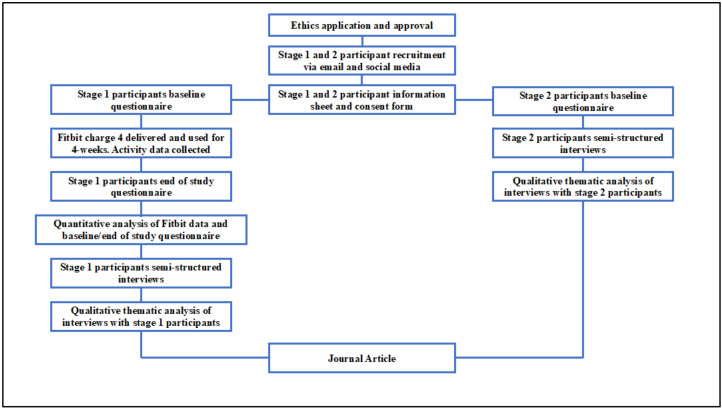
Flow diagram showing summary of study procedures for stage 1 and 2 participants.

**Figure 2 ijerph-18-11598-f002:**
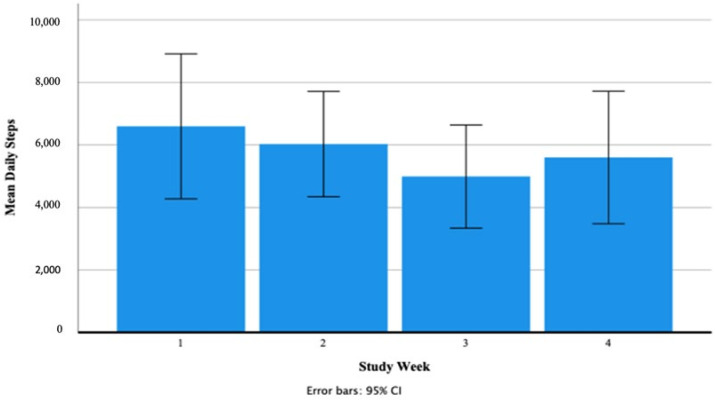
Participants’ mean daily steps as recorded via the Fitbit Charge 4 activity tracker.

**Figure 3 ijerph-18-11598-f003:**
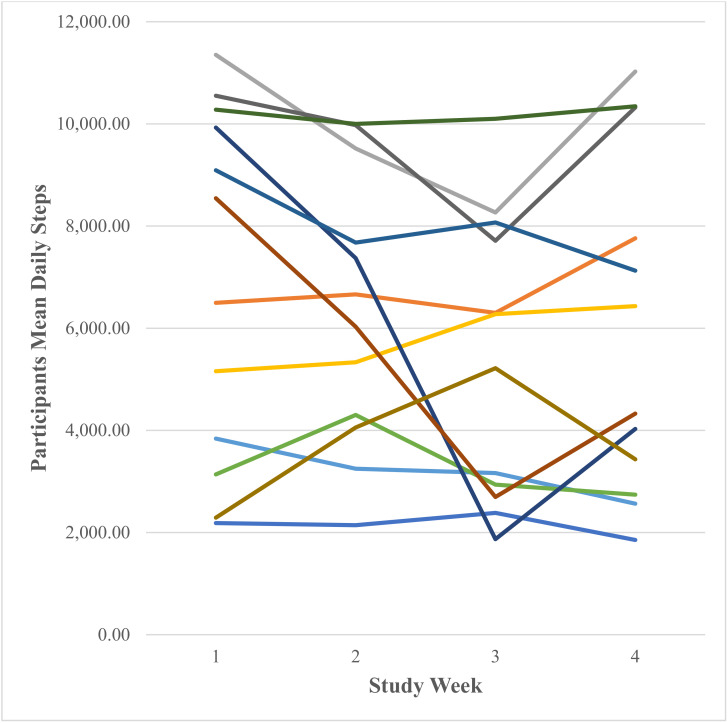
Individual participants’ mean daily steps as recorded via the Fitbit Charge 4 activity tracker (each coloured line represents one of the 12 participants).

**Figure 4 ijerph-18-11598-f004:**
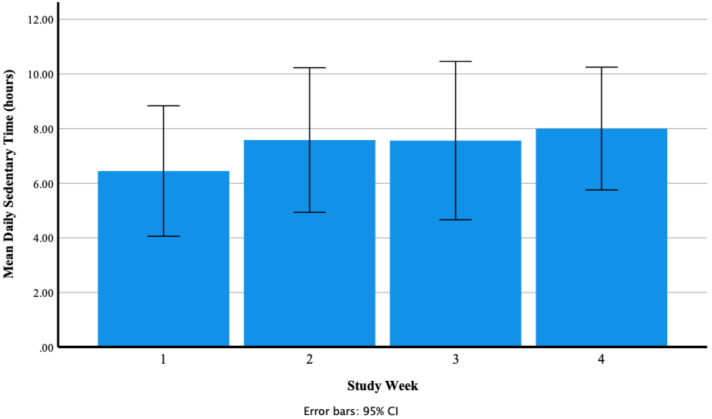
Participants’ mean daily sedentary time (hours) as recorded via the Fitbit Charge 4 activity tracker.

**Figure 5 ijerph-18-11598-f005:**
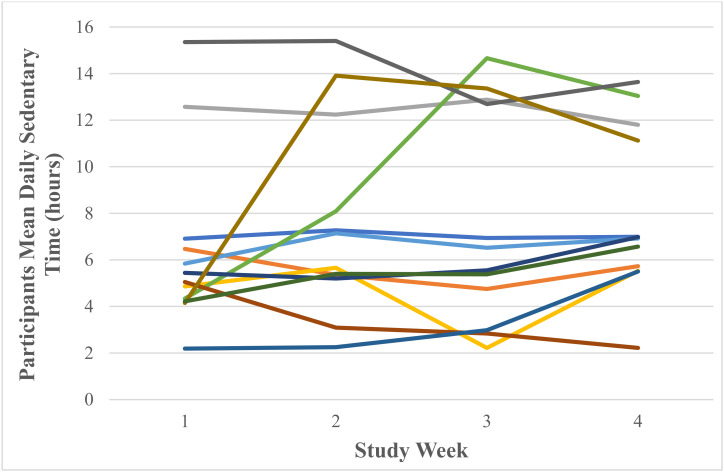
Individual participants’ mean daily sedentary time (hours) as recorded via the Fitbit Charge 4 activity tracker (each coloured line represents one of the 12 participants).

**Figure 6 ijerph-18-11598-f006:**
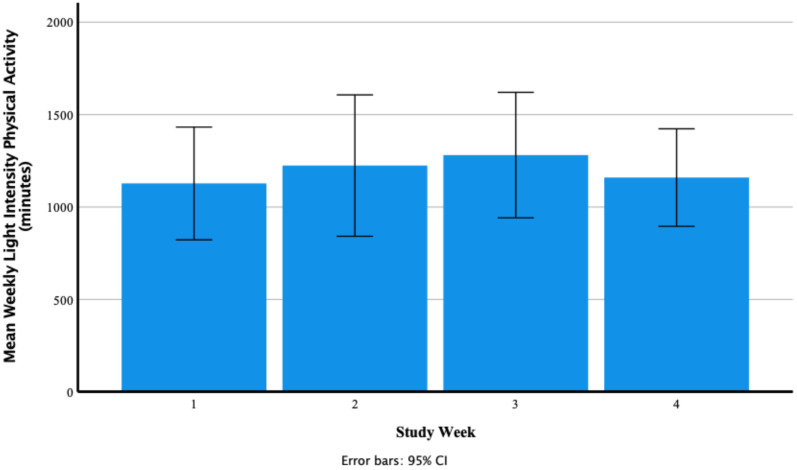
Participants’ mean weekly light intensity physical activity (minutes) as recorded via the Fitbit Charge 4 activity tracker.

**Figure 7 ijerph-18-11598-f007:**
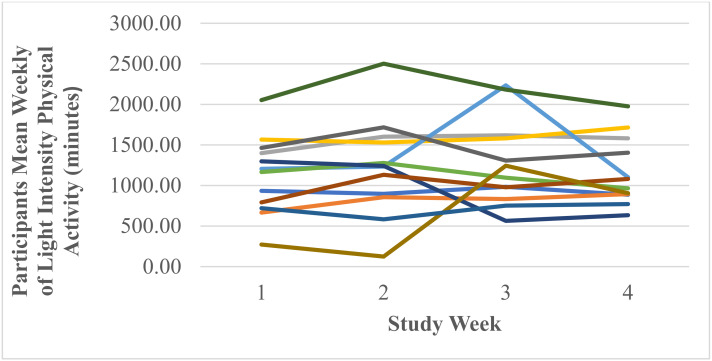
Individual participants’ mean weekly minutes of light intensity physical activity (each coloured line represents one of the 12 participants).

**Figure 8 ijerph-18-11598-f008:**
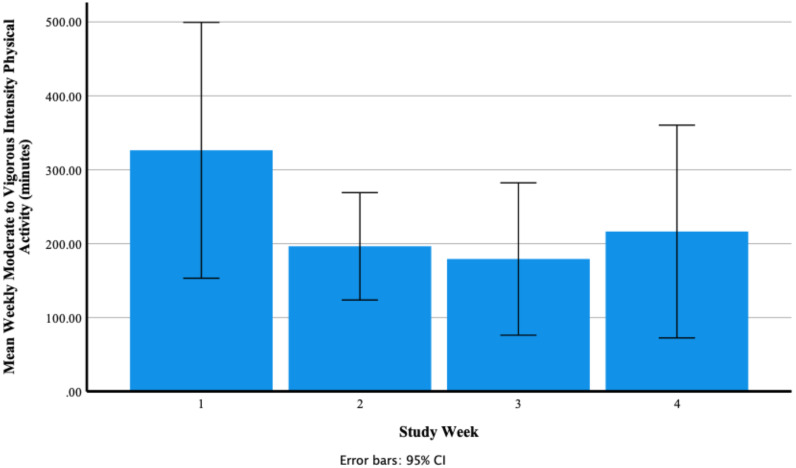
Participants’ mean weekly moderate- to vigorous-intensity physical activity (minutes) as recorded via the Fitbit Charge 4 activity tracker.

**Figure 9 ijerph-18-11598-f009:**
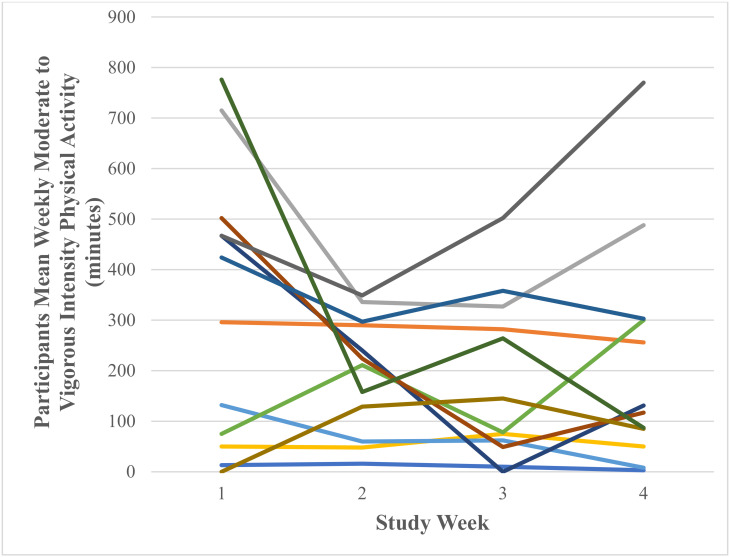
Individual participants’ mean weekly moderate- to vigorous-intensity physical activity (minutes) as recorded via the Fitbit Charge 4 activity tracker (each coloured line represents one of the 12 participants).

**Figure 10 ijerph-18-11598-f010:**
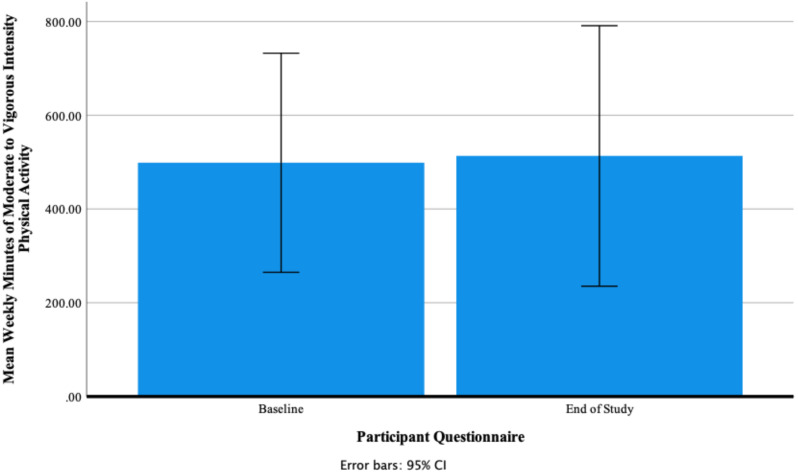
Participants’ self-reported mean weekly minutes of moderate- to vigorous-intensity physical activity at baseline and end of study.

**Figure 11 ijerph-18-11598-f011:**
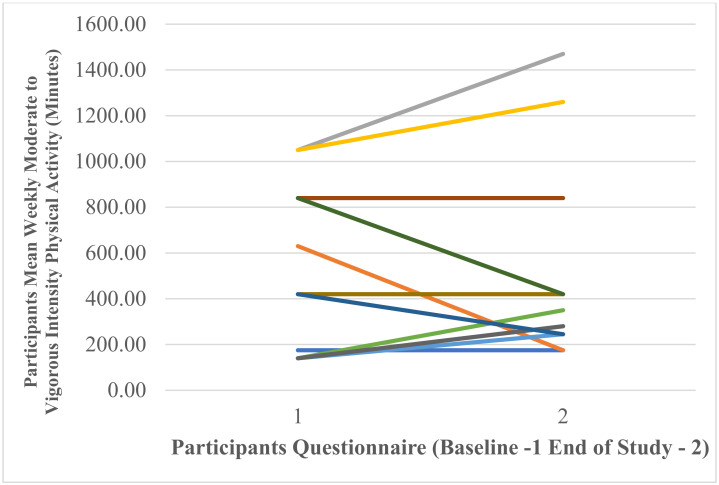
Individual participants’ self-reported mean weekly minutes of moderate- to vigorous-intensity physical activity at baseline and end of study (each coloured line represents one of the 12 participants).

**Figure 12 ijerph-18-11598-f012:**
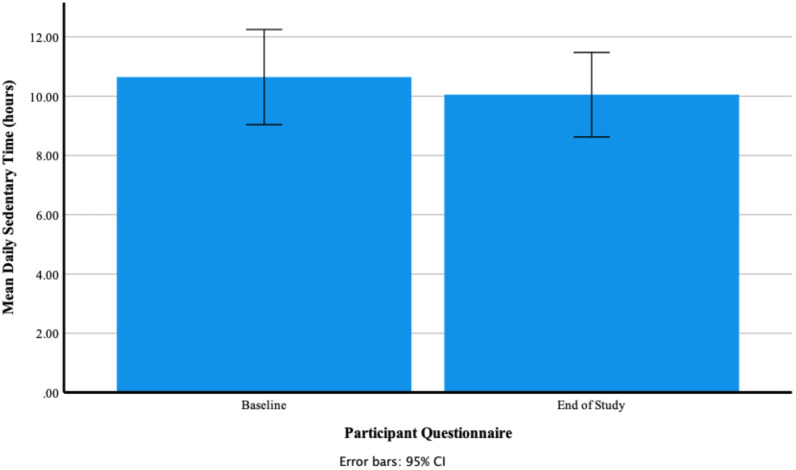
Participants’ self-reported mean daily hours of sedentary time at baseline and end of study.

**Figure 13 ijerph-18-11598-f013:**
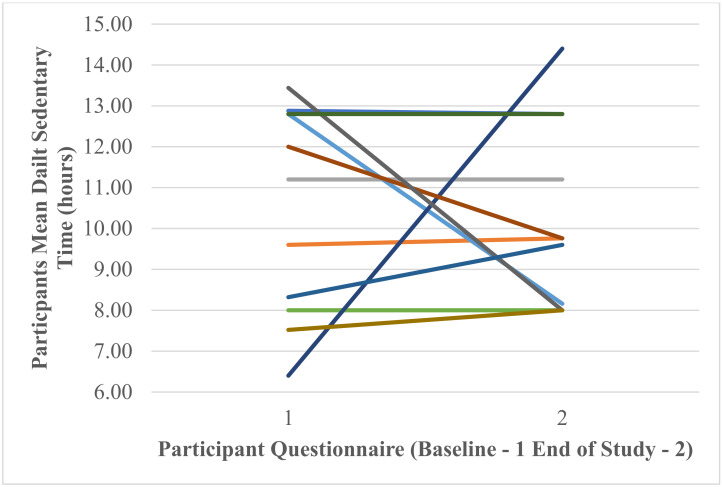
Individual participants’ self-reported mean daily hours of sedentary time at baseline and end of study (each coloured line represents one of the 12 participants).

**Figure 14 ijerph-18-11598-f014:**
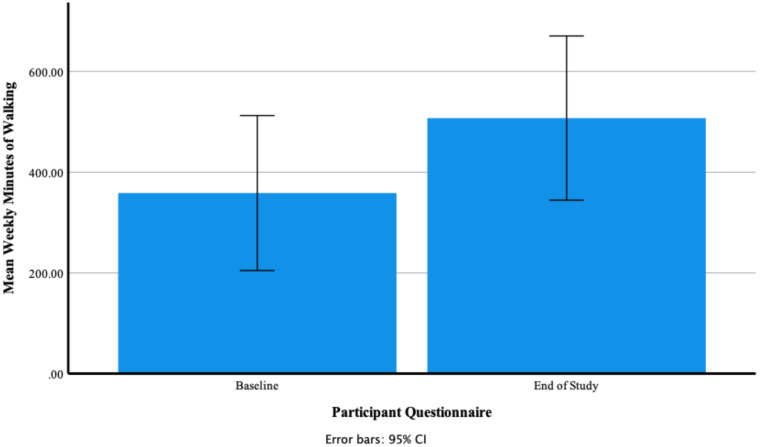
Participants’ self-reported mean weekly minutes of walking at baseline and end of study.

**Figure 15 ijerph-18-11598-f015:**
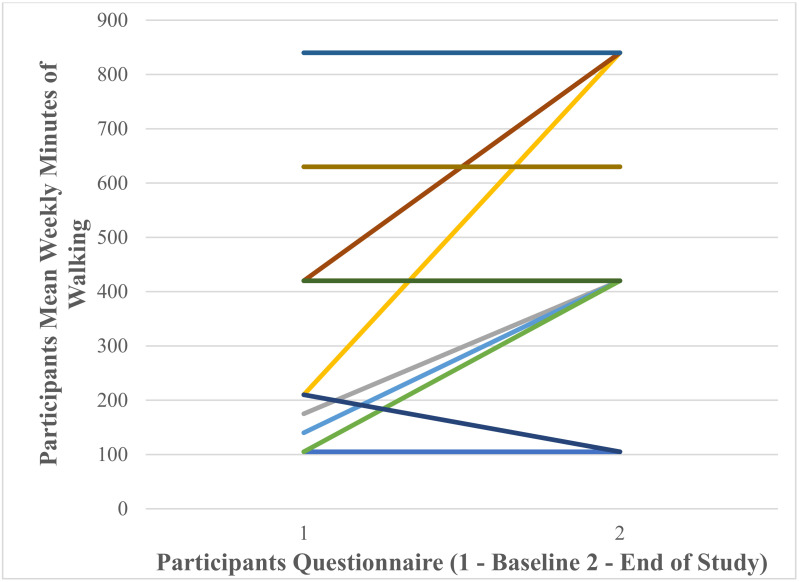
Individual participants’ self-reported mean weekly minutes of walking at baseline and end of study (each coloured line represents one of the 12 participants).

**Table 1 ijerph-18-11598-t001:** Demographic and individual profiles of stage 1 participants.

Participant Number	Gender	Age (Years)	Diabetes Duration (Years)	Weight (kg)	Height (m)	BMI	Blood Glucose (HbA1c %)
1	Male	77	9	104.0	1.88	29.5	7.8
2	Male	70	12	87.4	1.93	23.4	5.2
3	Male	52	6	100.0	1.93	26.8	7.2
4	Female	66	11	96.0	1.70	33.2	6.5
5	Male	66	30	103.0	1.83	30.8	5.4
6	Male	46	5	141.0	1.82	42.6	6.1
7	Female	57	1	78.0	1.60	30.5	6.0
8	Male	51	17	112.5	1.88	31.9	6.3
9	Female	56	7	87.0	1.60	34.0	7.9
10	Male	69	16	80.0	1.73	26.8	7.5
11	Male	53	2	97.0	1.83	29.0	12.0
12	Female	42	10	82.0	1.63	30.8	7.0
Mean		58.8	10.5	97.3	1.78	30.8	7.1

**Table 2 ijerph-18-11598-t002:** Identified main themes and associated sub-themes with the number of in-text references displayed in brackets.

Main Themes (*n* = 7)	Sub-Themes (*n* = 40)
**Current delivery of physical activity advice within type 2 diabetes health care (158)**	Consequences of a sedentary lifestyle (4). Current type 2 diabetes health care (18). Focus of current type 2 diabetes health care (14). Health care professional’s analysis of patient’s physical activity (5). Limitations of current type 2 diabetes health care (32). Patient’s physical activity at time of diagnosis (5). Impact of COVID-19 restrictions (5). Limitations of current type 2 diabetes advice (3). Limited physical activity advice (16). Method of current physical activity promotion (16). Current physical activity support (13). Health care professional promoting physical activity (13). Physical activity message (3). Regularity of physical activity advice (6). Sources of physical activity advice (2). Health care professional’s time management (3).
**Integrated elements of type 2 diabetes health care (14)**	Integration of Fitbit data with NHS information technology systems (5). Integrated type 2 diabetes health care (3). My Diabetes My Way data management (4). My Diabetes My Way data sharing (2).
**Data security and management (36)**	Data protection (8). Data security (8). Data sharing (11). Data use (9).
**Personalisation of type 2 diabetes physical activity support (45)**	Additional physical activity support (17). Personalisation of type 2 diabetes health care (19). Physical activity plan (3). Preferred type of physical activity (6).
**Use of Fitbit as a motivational and goal setting tool (48)**	Type 2 diabetes self-management (1). Fitbit as a motivational tool (20). Fitbit target group (2). Future use of mobile physical activity technology (5). Use of Fitbit as a goal setting tool (10). User analysis of Fitbit data (4). Users Fitbit experience (6).
**Users preferred Fitbit functions (57)**	Use of Fitbit data (20). Use of Fitbit functions (32). Popular Fitbit functions (5).
**Barriers to Fitbit use (7)**	Less popular Fitbit functions (3). Fitbit barriers (4).

**Table 3 ijerph-18-11598-t003:** Detailed analysis of stage 1 main themes.

Main Theme	Data Analysis
Current delivery of physical activity advice within type 2 diabetes health care	During interviews participants discussed how physical activity was currently promoted within their type 2 diabetes health care.
	The lack of physical activity support and the impact upon the health of participants was highlighted as a concern.
	*“If they had told me how much exercise impacted upon my blood glucose levels would have helped but they only skimmed over the subject. I have noticed that days when I don’t exercise my glucose levels rise”*
	The majority of participants stated that physical activity advice within their type 2 diabetes health care was limited. The advice was mainly promoted, by a health care professional, through a short discussion around present activity levels and the importance of being more active. Detailed analysis of patient’s activity was seldom undertaken, and follow-up exercise support was not offered.
	*“I have been diabetic for about 10 years and only had a limited amount of advice regarding physical activity”*
	*“In terms of when I was diagnosed with type 2 diabetes the only thing that I was told was to try and increase any level of activity”*
	Participants stated that the main focus of their type 2 diabetes care was around medication, diet and blood glucose levels. Physical activity was seldom discussed in detail and was not regularly promoted during visits to clinics.
	*“More time is spent discussing medication, blood glucose readings and diet”*
	*“The support needs to cover everything including activity, medication and diet”*
	During type 2 diabetes consultations with a health care professional detailed analysis of the patient’s physical activity was seldom undertaken. During these meetings the needs of patients and their preferred choice of exercise were never analysed in detail.
	*“They have never actually drilled down into how much of the time I am actually spending being active”*
	*“I do also have the occasional session with my doctors practice nurse. She has told me to keep active. Other than these short discussions I have received nor further advice regarding physical activity”*
	Two participants, who exercised on a regular basis and were knowledgeable of the benefits for their type 2 diabetes, found that physical activity advice available through different diabetes support sources conflicted. In their opinion such conflicting information caused confusion and would be more difficult to understand for those patients who were less active.
	*“There was even a conflicted with the advice that Diabetes UK and NHS gave especially in relation the physical activity”*
	Most of the participants said that physical activity advice was most commonly communicated verbally at only a superficial level by their health care professional. In the majority of cases exercise promotion was delivered by either a diabetes practice nurse or a clinic practice nurse.
	*“This has been verbally given by the practice nurse. Who is also a diabetic practice nurse”*
	*“The practice nurse did inform me verbally to be more active but that was just a short discussion”*
	Three participants declared that within their type 2 diabetes health care practice additional physical activity support was provided beyond just a verbal discussion. Two of the patients were offered free access to a local authority sports facility including exercise advice from a qualified personal trainer. One participant said that in their clinic a walking group had been established for those diagnosed with type 2 diabetes with the aim to increase levels of physical activity. Both of these options proved to be popular with patients.
	*“We were provided with free access to the community gym and health centre by the local nursing team”*
	*“I was also signed up for a diabetes walking support group through my GP …”*
	The regularity of physical activity promotion varied between participants. For some physical activity was discussed briefly during each 3–4 monthly diabetes clinic. For the remainder physical activity had seldom been discussed since first diagnosis 10–11 years earlier.
	*“I was diagnosed about 11 years ago and other than the occasional reminder to be active there has been no further advice or support”*
	*“I get a regular 4-month health check from a nurse in the local GP practice. She usually asks about my physical activity and depending on the circumstances she may nag a bit about you know you should be exercising”*
	A few participants thought that time pressures on health care staff was the main reason physical activity received only limited promotion. For staff to go beyond a brief discussion would require significantly more time and resources.
	*“…the staff just don’t have the time and their focus is more on my glucose levels and medication”*
Integrated elements of type 2 diabetes health care	Participants discussed the need to integrate all elements, including physical activity, of type 2 diabetes health care into a single support package.
	Five participants, who were registered with the MY Diabetes My Way programme, suggested that they would like to see their Fitbit data downloaded onto the system database and someone provide them with personalised physical activity analysis and feedback.
	*“It would be good if I could input my activity data into the My Diabetes My Way system and it gave me some feedback along the lines of well-done you are doing well or you need to do more of a specific activity”*
	*“This would allow for a far more personalised approach and probably get people more active”*
	The availability and role of type 2 diabetes health care support technology was mentioned by a few of the participants. In the experience of these patient’s available technology was only capable of undertaking a single specific task such blood glucose monitors. Participants desired for a single item of technology that could measure and provide feedback for blood glucose levels, physical activity and diet.
	*“It would be great having an all-round support package that combined both physical activity and diet. This is not available at the moment”*
	*“Another good thing would be the development of a combined activity and glucose monitor. I would find this very useful during my day-to-day life”*
Data security and management	All participants indicated that they would be happy to share their data from a Fitbit activity tracker with NHS information technology systems and directly with health care professionals. Participants trusted data security, storage and management of systems available within the health care sector. The use of Fitbit data was also supported by patients if it would lead to increased physical activity support.
	*“I have no worries about sharing my activity data with my doctor, nurse or on the My Diabetes My Way system”*
	*“I would be more than happy for my Fitbit data to be linked into a health care system especially if being used to support me”*
Personalisation of type 2 diabetes physical activity support	The majority of participants indicated that they would like to receive additional physical activity support from health care professionals that was personalised and focused on their exercise needs and lifestyle.
	*“The care team could spend more time going over this with me and make it more personalised”*
	Additional physical activity support from health care staff was desired by most of the participants. Suggested improvements included more discussion time, greater analysis of present activity levels and development of detailed exercise programmes.
	*“Developing a personalised training plan for users would be a great addition. This could be set around the users preferred choice of activity i.e., walking or cycling. This support could also include information on the benefits of physical activity for type 2 diabetes”*
	Physical activity support from non-health care sources was favoured by a few of the participants. Such support included family/friends, diabetes support groups and community-based walking groups.
	*“Yeah I think that I am quite fortunate that I have my husband and children to support me”*
	*“I would like to explore diabetes support groups where I could share my experiences with others”*
Use of Fitbit as a motivational and goal setting tool	Most participants highlighted the motivational and goal setting impact of the Fitbit Charge 4 activity tracker during their trial of the device over a 4 week period.
	The ability of participants to record and review their daily activity proved popular and motivated some to be more active.
	*“I was taking more steps than I thought, and this certainly motivated me to get out more”*
	*“The Fitbit has encouraged me to get out walking more which I have really enjoyed”*
	As some participants became more familiar with the Fitbit functions they started to develop their own physical activity goals. Reaching these goals produced a sense of achievement for users.
	*“I set my own little goals and the Fitbit was very good for doing this. Setting these goals and reaching them made me feel good about myself”*
	For half of participants the Fitbit allowed them to analysis their activities in more detail. Beyond the step counting function users found the heart rate monitor provided feedback on the intensity of their physical activity, which encouraged them to increase the pace during walks.
	*“Combing both of these functions was great for me as I could measure the number of steps but also see how hard I was working. In relation to the heart rate, I found this very reassuring to see how hard I was working”*
Users preferred Fitbit functions	All participants discussed their experience of using the Fitbit Charge 4 activity tracker and the functions they found useful in support of an active lifestyle.
	The most popular Fitbit function with most users was the step counter followed by the distance moved element. The step and distance functions were easy to understand without the need for further advice or support and complemented the most popular activity of walking.
	*“I found the steps counting function very good. I was able to see what I was doing when out for my walks”*
	Some users found the Fitbit heart rate monitor useful both from a novelty perspective and a tool to provide feedback on exercise intensity.
	*“In relation to the heart rate, I found this very reassuring to see how hard I was working”*
	During interviews some of the participants suggested that they would like a health care professional to incorporate the use of Fitbit functions within their type 2 diabetes treatment with a view to increasing activity levels.
	*“I think it would a good thing if my diabetes nurse could access my data and then we could sit done and chat about it and how to improve things or just keep me active”*
	A small number of participants found both the sleep and nutrition Fitbit functions useful though these did not form part of the study aims.
	*“I also liked the sleep function. This was good to see the amount and quality of my sleep”*
	*“The calories in and calories out was also interesting”*
Barriers to Fitbit use	Barriers to the use of the Fitbit Charge 4 activity tracker were raised by a small number of the participants. One participant initially had difficulty setting the device up and required support from a family member. Once setup, the participant became more confident in using this new technology. A second participant found the Fitbit too complicated to use without additional support.
	*“I wasn’t sure at the beginning. My husband had to set it up … but as I got used to it I really found it motivating”*
	*“I couldn’t understand some of the functions which were too complicated for me”*

**Table 4 ijerph-18-11598-t004:** Identified main themes and associated sub-themes with the number of in-text references displayed in brackets.

Main Themes (*n* = 6)	Sub-Themes (*n* = 32)
**Present promotion of physical activity within type 2 diabetes health care (151)**	**Limited physical activity promotion (16).** **Method of present physical activity promotion (13).** **Patient physical activity knowledge (15).** **Patient response to health care physical activity advice (14).** **Patient engagement with physical activity promotion (7).** **Physical activity guidelines (4).** **Physical activity promotion barriers (18).** **Present physical activity promotion (64).**
**Data security and management (8)**	**Fitbit data management and security (7).** **Patient data sharing consent (1).**
**Fitbit functionality (29)**	**Fitbit functions (21).** **Health care professional’s Fitbit knowledge (8).**
**Fitbit health care barriers (8)**	**Barriers to using Fitbit activity trackers within health care (7).** **Restrictions to using Fitbit within health care (1).**
**Future use of Fitbit within type 2 diabetes health care (93)**	**Fitbit behaviour change model (2).** **Fitbit data analysis software (3).** **Fitbit premium service (1).** **Fitbit social prescription (6).** **Future development of Fitbit functions (10).** **Future use of Fitbit data within patient health care (27).** **Use of Fitbit as a motivational and goal setting tool (4).** **Use of Fitbit to support physical activity promotion (19).** **Use of Fitbit within health care (19).** **Use of mobile applications to support the promotion of physical activity (2).**
**Improving physical activity promotion (74)**	**Integration of type 2 diabetes treatments (15).** **Focus of type 2 diabetes health care (12).** **Lifestyle medicine (17).** **Person responsible for future physical activity promotion (6).** **Physical activity training for health care professionals (3).** **Preventative health care promotion (3).** **Social group support for physical activity (5).** **Improvements in patient health care (13).**

**Table 5 ijerph-18-11598-t005:** Detailed analysis of stage 2 main themes.

Main Theme	Data Analysis
Present promotion of physical activity within type 2 diabetes health care	Participants provided feedback in relation to the present promotion of physical activity within type 2 diabetes health care. Medical staff in general stated that physical activity promotion within the NHS was limited and usually involved a short lifestyle discussion with the patient. Time constraints on health care staff was identified as a major barrier to promotion. Where physical activity advice was provided medical professionals found that this was ignored by patients and especially those already inactive. A few health care staff provided additional support in the form of self-help groups and targeted activity programmes. These sessions tended to be organised out with the normal working day of the clinician and were developed through a personal interest in type 2 diabetes. Generally, the promotion of physical activity is delivered by either a clinics practice nurse or a specialist diabetes nurse. The main focus of physical activity promotion is presently on prevention rather than treatment. Activity trackers are occasionally discussed with patients but never prescribed.
	*“We have nearly 800 diabetic patients under our care and it is impossible to see them in detail”*
	*“When you do give them advice they generally do not take it seriously”*
	*“One of my colleagues has started a diabetes project to support patients and part of this was running a support and walking group”*
	*“We just don’t have the time to spend on support packages. The promotion is usually a short lifestyle discussion with the patient”*
Data security and management	All health care and fitness professionals were satisfied that the security and management of Fitbit data shared with NHS information systems and medical staff would be used appropriately and stored securely. The need to obtain consent from the Fitbit user to store and use their activity data was recommended. Fitbit management participants provided reassurance that their company would not share user data unless consent is obtained from the device owner.
	*“As long as it was shared over a secure confidential system I cannot see this being an issue. At the end of the day the patient will make the final decision”*
	*“Overall security data is stored and managed safely within the NHS. The important element is patient consent”*
	*“If you are clear about what data you are collecting and why you are collecting that data there should be no issue. For patients they can opt in or out of data sharing”*
Fitbit functionality	Participants discussed the Fitbit functions and how these could be used to support an active lifestyle in people diagnosed with type 2 diabetes. The Fitbit step counting facility was highlighted as the most useful within health care and an easy concept for patients to understand. Health care staff also highlighted the heart rate, nutrition and sleep functions as valuable tools for type 2 diabetes diagnosis and treatment. Fitbit management participants suggested utilising the Fitbit communities function for the establishment of type 2 diabetes support groups and programmes. The communities function allows users to sign up for challenges and join or setup support groups. The Fitbit premium service (presently £7.99 per month or £79.99 per year) was also recommended by Fitbit management participants as a useful physical activity intervention package. The premium service includes personalised activity programmes, motivational games/challenges, in-depth activity and health analysis, exercise workouts and mindfulness support (breathing and relaxation exercises).
	*“… the step count is useful even just as a baseline. 10,000 steps is not achievable for everyone, but it is easier for me to set achievable step goals using these activity trackers”*
	*“For health care professional’s data such as heart rate and sleep patterns are useful. In relation to sleep there are studies that have shown that sleep can effect an individual’s glucose sensitivity. The heart rate is good for measuring a patient’s metabolic rate again an important factor in diabetes”*
	*“The Fitbit app has communities as part of the system. For example, we have just started a partnership programme with Diabetes UK which will hopefully help those that sign up to the specific Diabetes UK community group with help from those that are in the same position”*
	*“There is a Fitbit premium service which users pay for. This takes the users data and analyses it and then makes recommendations for the user. It is person centred”*
Fitbit health care barriers	Participants discussed the limitations and barriers to using Fitbit activity trackers within type 2 diabetes health care. The main identified barrier was cost and who should fund the purchase of a device. Medical participants highlighted the limited budgets available to treat patients and the additional financial burden that Fitbit activity trackers would place on resources. If patients were expected to purchase an activity tracker as part of their treatment this could exclude certain individuals from lower income families. A further barrier discussed was the information technology skills of patients which may be limited and require additional support including suitable internet access.
	*“The main negative would be the cost element and who covers this. Not all patients can afford these items”*
	*“I would love to have some to hand out, but our budget would not cover that”*
	*“I think these basic tools are good though they can cause stress for patients as well if they are not confident using technology”*
Future use of Fitbit within type 2 diabetes health care	The potential future use of Fitbit activity trackers within type 2 diabetes health care was examined by the interviewees. Medical professionals stated that activity trackers would be a valuable tool to examine patient’s physical activity levels and develop intervention programmes around their needs. Some suggested using Fitbit trackers as a cost-effective physical activity treatment rather than relying on more expensive medications. Fitbit management participants discussed future developments of the device including the ability to provide electrocardiogram readings, measurement of blood glucose levels and production of software capable of providing detailed physical activity analysis. Social prescription was also recommended with pharmacies being a hub for the distribution of activity trackers and provider of physical activity support. It was suggested that social prescription would reduce the pressures on frontline medical services.
	*“I would be interested in developing the use of trackers within my own health care. The feedback would a be a great tool when advising patients”*
	*“Data generated from a Fitbit is a prime example. Such data could be analysed by either a doctor or diabetic nurse and prescribed exercise promoted”*
	*“I know with some of the new products coming out that there is a greater focus on the heart and being able to produce ECG type feedback. We are also looking at the possibility of also developing devices that can measure insulin and glucose levels”*
	*“The ability to build a programme around social prescription that can be used by GP’s or increasingly through migration of services to pharmacies. Giving pharmacies more information and more tools to help them help the local community”*
Improving physical activity promotion	The improvement of physical activity promotion within type 2 diabetes health care was explored with participants. Better signposting of physical activity support within the NHS was suggested as this is presently an under used method of treatment. Dedicated physical activity trained staff was also recommended who could develop personalised intervention packages for patients. Further support in the form of specific activity sessions and exercise groups could be established within medical centres. The development and use of tools such as Fitbit’s and behavioural change interventions should be explored so as to allow patients to take greater control of their own lives. Increased use of physical activity prescriptions was suggested which allow health care staff to direct patients to local sports facilities at a reduced access cost.
	*“I think more sign posting direction patients to physical activity services”*
	*“The personalisation of promotion is really important, and this is not widely used within the NHS”*
	*“I expect the GP’s have more access to exercise prescription. For example, free gym or swimming passes”*
	*“Wellbeing groups are also good at supporting behaviour change”*

## Data Availability

All data is stored on the secure University of Strathclyde OneDrive system. This data can be made available by contacting the lead author.

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
