# Peer review of "Exploring the Use of Fitbit Consumer Activity Trackers to Support Active Lifestyles in Adults with Type 2 Diabetes: A Mixed-Methods Study"

_ijerph, 2021, doi:10.3390/ijerph182111598_

Round 1

Reviewer 1 Report

The issue raised is interesting, and the introduction of the Fitbit activity tracker to the treatment process has served its purpose as a motivator for taking up physical activity. The applied research program is well developed, but a drawback of the research is the small size of the research group.

With such a large difference in the age range of the respondents (21-58 years), we are dealing with a varied level of physical fitness. Instead of using only average results, individual profiles of patients should be shown

Information on the age and sex of the respondents should be provided in chapter 2.2. instead of lines 367-369. There is no information about the body weight of the respondents, which was certainly a factor that differentiated the movement abilities of the respondents.

Do the results in lines 344-347 refer to the same value ranges? The description of the data shows that the range of values increased at the end of the research.

Author Response

Date: 26 October 2021

Revisions Cover Letter for IJERPH – 1429368

Dear Sir/Madam

I refer to my proposed Journal Article submission IJERPH – 1429368 and your recommended revisions.

I would like to take the opportunity to thank the reviewers for their time and detailed recommendations for revision. Please find below revision tables with details of the changes made. In addition, the revisions are now tracked on the revised Journal Article.

Kind regards

William Hodgson

Reviewer Number 1

Reviewer Recommended Revision

Authors Response

1.     The applied research program is well developed, but a drawback of the research is the small size of the research group.

1.     This research program formed part of my MRes studies at the University of Strathclyde. The program ran for 1 year between 2020 and 2021. My thesis incorporated a literature review and a journal article. Due to the timescales involved my study protocol set the participant numbers as reported within this study. The small numbers of participants has been highlighted as a limitation within this journal article. I have just embarked on a 3-year PhD study in Physical Activity for Health which will expand on my MRes program with a view to increase participant numbers based on the findings within this article.

2.     With such a large difference in the age range of the respondents (21-58 years), we are dealing with a varied level of physical fitness. Instead of using only average results, individual profiles of patients should be shown.

Information on the age and sex of the respondents should be provided in chapter 2.2. instead of lines 367-369. There is no information about the body weight of the respondents, which was certainly a factor that differentiated the movement abilities of the respondents.

2.     Information on participants moved to chapter 2.2 (257 – 279) as recommended. Table 1 added with additional information relating to weight. Mean totals are listed at the bottom of this table.

3.     Do the results in lines 344-347 refer to the same value ranges? The description of the data shows that the range of values increased at the end of the research.

3.     The figures mentioned were the range of hours though the way this was written it indicated that self-reported sedentary time had increased. As this data is to some extent irrelevant the ranges have been removed (476 - 481).

Reviewer 2 Report

This is a well-written manuscript that addresses a really interesting topic. However, I have several major concerns:

  1. Abstract. The Results sections of the Abstrct is poor. Some figures should appear regarding the quantitative part of the study
  2. Introduction. An important citation in the field of Diabetes is missing regarding the premature mortality of such disease (Baena-Diez JM. Diabetes Care. 2016)
  3. The objective is very generic: "The aim of this study was to explore the use of Fitbit consumer activity trackers to support active lifestyles in adults with T2D".   In which sense did the authors explore the use of Fitbit in T2D patients?
  4. Recent experiences in Mixed Methods studies should be cited in Discussion (Diaz JL.  Inform Health Soc Care. 2021)
  5. Please, be consistent with the use of abbreviations. Choose between T2D or type 2 diabetes 

Author Response

Date: 26 October 2021

Revisions Cover Letter for IJERPH – 1429368

Dear Sir/Madam

I refer to my proposed Journal Article submission IJERPH – 1429368 and your recommended revisions.

I would like to take the opportunity to thank the reviewers for their time and detailed recommendations for revision. Please find below revision tables with details of the changes made. In addition, the revisions are now tracked on the revised Journal Article.

Kind regards

William Hodgson

Reviewer Number 2

Reviewer Recommended Revision

Authors Response

  1. Abstract. The Results sections of the Abstract is poor. Some figures should appear regarding the quantitative part of the study.

1.     As recommended the results section of the abstract has been amended. It now incorporates quantitative data relating to the self-reported information provided by the stage 1 participants 18 – 29).

  1. Introduction. An important citation in the field of Diabetes is missing regarding the premature mortality of such disease (Baena-Diez JM. Diabetes Care. 2016).

2.     At 42 – 73 an additional citation discussing the hazard risk of cause specific death as reported by Baena-Diez et al. 2016 has been added as recommended.

  1. The objective is very generic: "The aim of this study was to explore the use of Fitbit consumer activity trackers to support active lifestyles in adults with T2D".   In which sense did the authors explore the use of Fitbit in T2D patients?

3.     At 214 – 216 the aim of this study was expanded to include the quantitative and qualitative elements of the project.

  1. Recent experiences in Mixed Methods studies should be cited in Discussion (Diaz JL.  Inform Health Soc Care. 2021)

4.     As recommended an additional citation has been included at 550 – 554 in relation to the experiences of Diaz et al. 2021 in respect of mixed methods studies.

  1. Please, be consistent with the use of abbreviations. Choose between T2D or type 2 diabetes 

5.     Recommendation noted and type 2 diabetes used throughout amended article rather than T2D.

Round 2

Reviewer 2 Report

I have no further comments